# Electroencephalographic Characterization of Sensorimotor Neural Activation During Swallowing in Dysphagic Patients

**DOI:** 10.3390/s25216767

**Published:** 2025-11-05

**Authors:** Javier Imaz-Higuera, Javier Garcia-Casado, Yiyao Ye-Lin, Jose Luis Martinez-de-Juan, Marta Gutierrez-Delgado, Jennifer Prieto-House, Gemma Mas-Sese, Araceli Belda-Calabuig, Gema Prats-Boluda

**Affiliations:** 1Centro de Investigación e Innovación en Bioingeniería (Ci2b), Universitat Politècnica de València, 46022 Valencia, Spain; jimahig@etsii.upv.es (J.I.-H.); yiye@ci2b.upv.es (Y.Y.-L.); jlmartinez@eln.upv.es (J.L.M.-d.-J.); gprats@ci2b.upv.es (G.P.-B.); 2BJUT-UPV Joint Research Laboratory in Biomedical Engineering, Universitat Politècnica de València, 46022 Valencia, Spain; 3Medical Rehabilitation Department, Medium- and Long-Term Care of Hospital Pare Jofré, 46017 Valencia, Spain; 4Medical Rehabilitation Department, Hospital La Pedrera, 03700 Alicante, Spain

**Keywords:** neurogenic dysphagia, electroencephalography (EEG), event-related spectral perturbation (ERSP), stroke, neurophysiology

## Abstract

**Highlights:**

**What are the main findings?**
EEG analysis revealed reduced beta desynchronization in motor cortical areas of post-stroke patients.Dysphagic patients showed specific deficits in alpha/beta desynchronization at C3 and Cz and exhibited abnormal lateralization.

**What are the implications of the main findings?**
EEG-derived biomarkers allow objective discrimination between dysphagic and non-dysphagic populations.The accessibility of EEG supports its integration into clinical workflows for dysphagia diagnosis and follow-up.

**Abstract:**

Dysphagia is commonly assessed with qualitative and image-based diagnostic tools, which are often costly, technically demanding, and limited in their ability to support individualized rehabilitation. Electroencephalography (EEG) has recently emerged as a quantitative, cost-effective, and accessible alternative to characterize sensorimotor activity during swallowing, though its potential in dysphagic populations has not been systematically explored. This study investigated neural dynamics in 50 post-stroke dysphagic patients, 32 post-stroke non-dysphagic controls, and 21 healthy adults performing a swallowing task. EEG recordings from primary motor regions (C3, Cz, C4) were analyzed using event-related spectral perturbation (ERSP) to quantify alpha (8–13 Hz) and beta (15–30 Hz) event-related desynchronization, alongside hemispheric lateralization indices. Group comparisons revealed significantly reduced beta desynchronization in both post-stroke groups compared to healthy participants, with additional alpha and beta deficits at C3 and Cz distinguishing dysphagic patients from non-dysphagic controls. Dysphagic patients further exhibited abnormal lateralization not observed in other groups. These findings identify distinct alterations in motor cortical dynamics and hemispheric balance in dysphagia, supporting EEG-derived biomarkers as promising tools for diagnosis and clinical follow-up. The accessibility of EEG reinforces its potential integration into routine workflows to enable objective and personalized management of post-stroke dysphagia.

## 1. Introduction

Dysphagia is a condition characterized by an impairment in the swallowing mechanism, resulting in difficulty with the proper ingestion of food and liquids. This physiological function can become a significant clinical concern, leading to malnutrition, dehydration, aspiration, pneumonia, reduced quality of life, social isolation, and increased healthcare costs [1,2,3]. While the causes of this condition are varied and differ across age groups, neurological origins are the most common in individuals over 60 years of age [4]. Neurogenic dysphagia affects between 43.6% and 58.8% of patients following a stroke [5], making it one of the most serious complications associated with this condition. Post-stroke patients have approximately 4 times the risk of aspiration pneumonia and a markedly higher mortality rate, and it is one of the most dangerous sequelae of this medical condition [6]. This condition places an additional burden on healthcare systems; for instance, in Denmark, the average cost of caring for a stroke patient during the first year is €27,711, with post-stroke dysphagia increasing this cost by approximately 23% [7]. Given the aging population and the rising prevalence of neurological disorders, these figures are projected to increase in the near future.

Standard screening techniques for dysphagia offer good diagnostic performance at a qualitative level, with the Volume-Viscosity Swallow Test (V-VST) being among the most commonly used [8]. Nevertheless, instrumental assessments are preferred, with videofluoroscopy (VFS) considered the gold standard, followed by fiberoptic endoscopic evaluation of swallowing (FEES) [9]. However, these methods present notable limitations, including exposure to ionizing radiation, potential allergic reactions to contrast agents in VFS, the requirement for highly trained personnel and specialized equipment, and the risk of mucosal injury during FEES [10]. Comprehensive reviews report detection rates of dysphagia ranging from 37% to 55% when using clinical screening alone, compared to 64% to 78% when instrumental methods are employed [11], indicating a high likelihood of underdiagnosis. These findings underscore the need for diagnostic tools capable of providing more objective and quantitative assessments.

The swallowing control system involves both the central and peripheral nervous systems. The cortex initiates and continuously coordinates swallowing, adjusting the activity based on feedback. The brainstem processes and integrates this feedback through motor and sensory pathways, ensuring smooth and efficient swallowing. The first studies of swallowing brain control involved imaging techniques to highlight the activation of the primary cortex, the sensorimotor cortex, and the premotor/supplementary areas during swallowing [12,13,14]. However, neurological control regarding the strength and coordination of muscular activity is barely understood. This gap in knowledge is particularly important given that the majority of swallowing disorders are caused by neurological impairments rather than issues with the peripheral structures involved in swallowing [15]. To investigate the uncompromised temporal patterns of swallowing in healthy individuals, several magnetoencephalography (MEG) studies have provided time-sensitive analyses of cortical sensorimotor responses, aligned with the initiation and termination of submental muscle activity as recorded through surface electromyography (sEMG) [16,17]. Dziewas et al. found bilateral activity during the performance of volitional swallowing, with dominance of the left hemisphere [16]. Subsequently, Teismann and colleagues observed a transition in hemispheric dominance from the left to the right hemisphere as swallowing advanced from the initial oral phase to the subsequent pharyngeal and esophageal phases [18]. Like MEG, electroencephalography (EEG) provides high temporal resolution for assessing the timing of neural activity and is well-suited for clinical use in evaluating swallowing dysfunction in patients with neurological conditions. Event-Related Spectral Perturbation (ERSP) is a well-established method used to characterize event-related changes in the spectral power of EEG signals over time, relative to a pre-event baseline. It provides a comprehensive view of both power increases (event-related synchronization, ERS) and power decreases (event-related desynchronization, ERD) across frequency bands and time windows. In this framework, ERD can be interpreted as a specific manifestation of ERSP, corresponding to sustained decreases in spectral power within a defined frequency band and time interval, typically visualized as negative deviations in the ERSP map. Cuellar et al. (2016) analysis [15] showed event-related desynchronization (ERD) in the spectral power in the alpha (8–13 Hz) and beta (15–25 Hz) frequency bands of the mu clusters during healthy swallowing. Furthermore, Koganemaru’s research [19] combined ERSP with corticomuscular coherence, finding desynchronized activity and oscillatory interaction between the cortex and pharyngeal muscles in the bilateral sensorimotor, premotor, and inferior prefrontal areas during volitional swallow in healthy humans. These studies collectively propose EEG as a valuable tool for mapping normal physiological swallowing, offering insights into both neural and muscular coordination during swallowing. However, to date, no studies have specifically focused on EEG alterations in individuals with post-stroke dysphagia.

This article provides the first ERSP characterization of EEG signals to suggest biomarkers related to the strength and lateralization of the activity for post-stroke dysphagia. Furthermore, we compare it not only with that of healthy subjects but also with that of post-stroke subjects without dysphagia. The results support further use of EEG ERSP analysis to provide quantitative biomarkers that can aid in post-stroke dysphagia diagnosis and could be used for personalized and effective treatment assessment.

## 2. Materials and Methods

### 2.1. Subjects

A total of 103 subjects were enrolled in this study: 21 healthy individuals (mean age 59.9 ± 6.3 years) with no history of dysphagia or related dysfunctions (healthy group); 32 stroke patients without dysphagia (mean age 67.1 ± 11.4 years, control group); and 50 stroke patients with post-stroke dysphagia (mean age 69.0 ± 12.9 years, dysphagic group). Further information about relevant demographic variables from the different study groups can be found in Table 1. Inclusion criteria for patient enrollment were age between 50 and 85 years, the ability to perform the tasks required for the Volume-Viscosity Swallow Test (V-VST), and a Charlson Comorbidity Index score of ≤3. The study protocol was approved by the Ethics Committees of La Pedrera (Ref. HPL_70_2021) and Pare Jofré (Ref. 1357) Hospitals, and all participants provided written informed consent.

Stroke diagnosis was confirmed through magnetic resonance imaging (MRI) or cranial computed tomography (CT), while the presence of dysphagia was consistently confirmed using the V-VST. In cases where results were inconclusive or further confirmation was deemed necessary, additional instrumental assessments such as VFS or FEES were conducted. These assessments were conducted following patient admission and prior to the initiation of any treatment, during the subacute phase of stroke, in order to evaluate the initial clinical condition.

### 2.2. Protocol and Data Acquisition System

Each participant underwent an EEG recording session while performing a standardized swallowing protocol. Subjects were seated comfortably in an examination chair and instructed to maintain a natural, upright head position (Figure 1). EEG signals were recorded using a 10–20 configuration with an electrode cap (TMSi—BrainWave cap, Oldenzaal, The Netherlands). An additional Ag/AgCl electrode was placed on the left mastoid to serve as a reference. Before electrode placement, the scalp was carefully prepared to reduce electrode–skin impedance to below 10 kΩ. Data were acquired at a sampling rate of 2 kHz using a commercial bioamplifier (TMSi SAGA 32+, Oldenzaal, The Netherlands).

The swallowing protocol is illustrated in Figure 2. Each trial began with the administration of a 5 mL bolus of thickened pineapple juice, delivered by syringe with minimal disturbance by the examiner. Participants followed visual cues displayed on a screen: a cross indicated the need to prepare for swallowing, and the command to swallow was given via a green screen with a red central circle (t = 0 in subsequent analysis). The protocol was repeated up to 40 times per subject, with termination if any signs of aspiration risk or patient discomfort were observed [8]. Only participants who completed a minimum of 20 valid swallows were included in the final analysis.

### 2.3. Processing

Figure 3 provides a schematic representation of the pipeline of EEG signal preprocessing and subsequent analysis, characterization, and statistical analysis performed.

EEG signal processing involved initial downsampling to 512 Hz, followed by the application of a high-pass filter at 7 Hz to remove motion artifacts, baseline drift, and attenuate slow EEG waves that were too artifactual for this study, like Cuellar et al. did [15]. Artifact Subspace Reconstruction (ASR) was used to eliminate channels with artifacts and to remove remaining motion artifacts in the signals with a threshold set at 10 standard deviations [20,21]. After removing the artifacts, the channels were reconstructed to the original 32-channel design by interpolation of the nearest ones. A low-pass filter at 35 Hz was applied to attenuate power line interference, muscular interference, and high-frequency noise. The data were then segmented into epochs from −1000 ms to 5000 ms relative to the swallowing cue, and the EEG channels were re-referenced by applying a Common Average Reference.

Independent Component Analysis (ICA) was performed over the epoched signals using the Picard algorithm in EEGLAB v2023.0 [20], which unmixes the channel recordings into statistically independent component processes. This helped capture anatomically and functionally distinct brain source processes and separate non-brain artifacts. Picard v1.0 is known for its fast convergence and efficiency compared with other algorithms like FastICA and Infomax [22]. ICLabel v1.6, an automated IC classifier, was then used to classify the independent components into broad source categories [23]. Components identified as muscular, eye movement, and non-cerebral were eliminated, and the remaining components were used to reconstruct the EEG signals. This method effectively removes artifacts while preserving information from most channels.

### 2.4. Data Analysis

The ERSP analysis in the 7 Hz to 30 Hz range was performed to assess changes in spectral power across time. These time–frequency transformations across trials (−1000 ms to 5000 ms) were computed using a Morlet sinusoidal wavelet transformation initially set at 3 cycles and rising linearly to 20 cycles at 30 Hz [24].

Baseline correction (over the entire epoch length) was applied to each trial ERSP individually to reduce the impact of large artifacts and the influence of outliers on the statistical analysis [25].

Subsequently, classical pre-stimulus baseline corrections were applied to the resulting ERSP trial averages. Event-related power at each time–frequency point is divided by the average spectral power, for each frequency bin, in the pre-stimulus baseline period [−1000 s to 0 s] at the same frequency. The log-transformed measure is derived by taking the log value of ERSP [25].

Only the electrodes C3, Cz, and C4, placed in the sensorimotor area, were studied in this work because they are thought to be involved in the initiation and control of the swallowing motor pattern, particularly during the pharyngeal phase, where precise coordination of muscles is essential to ensure safe and efficient swallowing [15,26,27].

To better describe the EEG activity in the alpha (7–13 Hz) and beta (13–30 Hz) frequency bands, we computed the mean spectral power over these ERSP bands across the entire time window (−1000 ms to 5000 ms). We called it the event-related oscillation (*ERO*).(1)$$ERO\;\left(t\right)=\frac{1}{f_{2}-f_{1}}\int_{f_{1}}^{f_{2}}ERSP\left(t,f\right)\cdot df$$
where $f_{1}$ and $f_{2}$ are the lower and upper limits of the frequency band considered. This facilitates the visualization of the level of ERD/ERS in each band over time during the swallowing process.

To further analyze the differences observed in the ERSP data, we obtained the mean spectral power changes over the time window from $t_{1}=\;500\;ms$ to $t_{2}=\;2000\;ms$, based on the swallowing times described in previous EEG and EMG studies [15,19,28,29]. This integration was performed for both the alpha ($f_{1}=\;7\;Hz$, $f_{2}=\;13\;Hz$) and beta ($f_{1}=\;13\;Hz$, $f_{2}=\;30\;Hz$) frequency bands across the C3, Cz, and C4 channels. We referred to it as the event-related index (*ERI*), calculated as follows:(2)$$ERI=\frac{1}{t_{2}-t_{1}}\frac{1}{f_{2}-f_{1}}\int_{t_{2}}^{t_{1}}\int_{f_{2}}^{f_{1}}ERSP\left(t,f\right)dfdt$$

A negative value of the ERI is equivalent to a desynchronization, and a positive value to a synchronization of brain activity.

Hemispheric lateralization concerning the swallowing-related activation was quantified using a lateralization index (LI), which was calculated as follows:(3)$$LI\;=\;\;\left|\frac{ERI_{C3}\;\;-\;\;ERI_{C4}}{ERI_{C3}\;\;+\;\;ERI_{C4}}\right|$$

The C3 and C4 correspond to the left and right hemispheres, respectively. LI with a value of 0 represents bilateral dominance [16,30].

The systematic lateralization of the activity towards one side was tested by comparing the ERI between C3–C4 electrode pairs within each subject group.

The ERI and the lateralization index were subjected to the Lillie and Levene tests, respectively, in order to ascertain whether normality and homoscedasticity were present. The results indicated abnormality and heteroscedasticity. A non-parametric approach was adopted to assess differences in the parameters across groups. This involved the use of the Kruskal–Wallis test, followed by post hoc comparisons using the Wilcoxon unpaired test (Mann–Whitney U test). The effect size was calculated for significant unpaired differences with the Cliff’s delta (δ). For the paired comparison between subjects’ electrodes, the Wilcoxon signed-rank test was used, and the effect size was checked with the rank biserial correlation (r). A significance level of *p* < 0.05 was considered for all the statistical tests performed. Given the exploratory nature of this study, correction for multiple comparisons was not applied, as the primary aim was to identify potential differences and trends that warrant further investigation in larger, hypothesis-driven studies. All the analyses performed were implemented using MATLAB’s Statistics and Machine Learning Toolbox v.24.2 [31].

## 3. Results

Figure 4 shows the mean ERSP across the C3, Cz, and C4 EEG channels for the three population groups under investigation. Event-Related Desynchronization (ERD), characterized by negative ERSP values (blue regions), is evident within the alpha frequency range (7 Hz to 13 Hz) and the beta range (13 Hz to 25 Hz). High-frequency beta desynchronization (>25 Hz) is exclusively observed in healthy individuals across all examined channels.

In the dysphagic group, the magnitude of the ERD is significantly diminished compared to the other groups (Figure 4). Moreover, within this group, the C4 channel exhibits a more pronounced ERD relative to C3, while the Cz channel displays only minimal desynchronization. Notably, in both the health and control groups, ERD is more intense during the initial two seconds than at later times. Dysphagic patients generally exhibited weaker alpha ERD, especially around the 500 ms to 2000 ms time window. This is depicted in more detail in the representation of the quantitative parameters shown in Figure 4 and Figure 5.

In all groups and channels, the ERD extends up to four seconds in both the alpha and beta bands, with the sole exception of the alpha band in the Cz channel of the healthy group, where the duration is shorter.

Figure 5 displays that dysphagic patients generally exhibit weaker alpha ERD, especially around the 500 ms to 2000 ms time window for C3 and Cz electrodes. This can be observed in more detail in the representation of the quantitative parameters shown in Figure 6. Notably, control subjects present very similar ERO evolution and values to healthy subjects, indicating minimal differences between these two groups in the alpha frequency band. Finally, it is also worth noting that in dysphagic patients, the ERD intensity for C3 is weaker than for C4, which is summarized quantitatively in the LI parameter shown in Figure 7. The bottom row of Figure 5(B1–B3) represents ERO evolution in the beta frequency band at the same electrode positions. Again, the ERD values of dysphagic patients are weaker than those of the other groups, with the most pronounced differences in the 500 to 2000 ms time interval. However, in contrast to the alpha band, the ERDs in the beta band of the control group are markedly weaker than those of the healthy group.

Figure 6 presents box–whisker plots, with the results of ERI for C3, Cz, and C4 in alpha and beta frequency bands for the three study groups. Significant differences in ERD values were found for all channels and frequency bands between the groups (Kruskal–Wallis: *p* < 0.05). Post hoc analysis proved that the weakest ERD values observed in dysphagic patients were significantly different from healthy (*p* < 0.05, in all channels and both bands, but alpha Cz) and from control at both bands in C3 (*p* < 0.001, δ > 0.42) and Cz (*p* < 0.019, δ > 0.32). The control group was only different from healthy at the beta band in all channels (*p* < 0.03, δ > 0.37). The C3 channel is of particular interest due to its significantly low ERIs, both for the alpha band (Median values: Dysphagic −0.31 dB, Control: −0.86 dB, Healthy: −1.80 dB) and beta band (Median values: Dysphagic −0.36 dB, Control: −0.91 dB, Healthy: −2.30 dB). It is also noteworthy that when comparing the control and healthy groups, no significant differences were found in the alpha band for any of the study channels, while significant differences (*p* < 0.05) were found in the beta band for all three channels.

As shown in Figure 7, the dysphagia group had the highest lateralization values for both frequency bands. This difference was statistically significant when compared to the healthy group for both alpha and beta, and also to the control group for the beta band. There were no significant differences between the control and the healthy group in any of the frequency bands.

Table 2 presents the statistical comparison (*p*-value) and effect size (r) for ERI paired electrode comparisons within each group and frequency band. C3 and C4 revealed no statistically significant differences in the alpha and beta bands for the dysphagic group, but it did for the control group in the beta band.

## 4. Discussion

Cuellar et al. [15] and Koganemaru et al. [19] mapped the time–frequency dynamics of sensorimotor activation relative to the swallow in healthy subjects, whose ERSP during swallowing showed a clear desynchronization of the sensorimotor areas in both the alpha and beta frequency bands. In the present work, a similar desynchronization pattern was obtained in the control and healthy groups, whereas the dysphagic group presented different characteristics, especially with a reduction in the ERSP desynchronization. As far as we are concerned, this work is the first to map time–frequency dynamics of sensorimotor activation relative to swallowing in dysphagic patients and stroke patients without dysphagia with EEG.

When comparing the ERD intensity of the different groups of subjects, it was observed to be significantly weaker in stroke patients with dysphagia than in healthy and control subjects in both alpha and beta bands. These bands are the main ones responsible for the swallow control [15,19,26]. ERD, in essence, represents a decrease in the synchronization of brainwave activity within a specific frequency band during the performance of a task or in response to a stimulus. This decrease in synchrony, counterintuitively, suggests that neurons are potentially more actively engaged in processing information related to the task. Therefore, a smaller or reduced ERD, as seen in the dysphagic group, can be interpreted as an indicator of neuronal hypoactivation in the sensorimotor cortical areas associated with swallowing. This aligns with some MEG studies that have suggested that hypoactivation of cortical regions associated with swallowing tasks after stroke is the underlying cause of swallowing disorders [32]. In the present work, no significant differences were found between the control and the healthy groups at the alpha frequency band. This has never been proven in studies related to swallowing; however, some have examined EEG dynamics in post-stroke rehabilitation patients and reported the preservation of alpha activity [33,34,35]. Stępień et al. [33] presented a preserved alpha ERD in non-paretic hand movements. Vico Fallani et al. [35] revealed the connectivity networks of finger tapping activity in stroke patients, and proved that the alpha connectivity network did not show a decrease in global and local efficiency in the patient’s networks, but the beta and gamma bands did. Carino-Escobar et al. [34] found a moderate relationship between alpha rhythms in frontal, temporal, and parietal areas with upper-limb motor recovery, which could imply a stronger relationship between non-dysphagic stroke impairment and the affected beta brain activity. These results align with our observations of preserved alpha desynchronization in post-stroke non-dysphagic (control) patients and the significant reduction in the alpha activity for the patients affected by dysphagia (*p* < 0.026, δ > 0.34 at every channel). On the other hand, the ERD in the beta band was significantly stronger for the healthy group in comparison to both post-stroke dysphagic and non-dysphagic groups. The beta band oscillations in the sensorimotor area are associated with the planning and preparation of the movement, as discussed by Cuellar et al. [15]. This suggests that the stroke affects the beta band regardless of dysphagia. However, it is important to note that the differences between healthy and dysphagic patients were greater than those between healthy and control groups. These results are similar to the MEG study of Teismann et al. [32], whose post-stroke non-dysphagic patients showed reduced ERSP in the beta band.

This study’s key contribution lies in identifying significant differences in Event-Related Spectral Perturbation (ERSP) patterns between post-stroke individuals with and without dysphagia. Importantly, to the best of our knowledge, this study is the first to undertake a comparative analysis of these two groups with EEG signals. The significant differences presented in ERSP-derived biomarkers indicate a specific neurological signature for dysphagia beyond general swallowing difficulty. The preservation of alpha desynchronization observed in the control group did not occur in dysphagic patients, whose desynchronization was weaker, and without a specific time interval of maximum ERD. Significant differences from the control group were found in the left (*p* = 0.001, δ = 0.42) and central (*p* = 0.002, δ = 0.4) sensorimotor areas during swallowing (~500 ms to 2000 ms). Alpha oscillation is thought to capture somatosensory information. Specifically, Cuellar et al. suggested that alpha ERD during swallowing may reflect the somatosensory feedback from sensory regions to the premotor cortex to update motor commands and guide adjustments in movement parameters [15]. This is probably the characteristic most affected in dysphagic patients, and interestingly, it starts at the beginning of the oropharyngeal phase (500 ms) [36]. The beta desynchronization was also significantly reduced in dysphagic patients compared to controls in the left (*p* < 0.001, δ = 0.47) and the central (*p* = 0.019, δ = 0.32) sensorimotor areas. These findings indicate a substantial impact on the planning and preparatory processes underlying motor execution [15]. Further studies could be proposed in this matter, principally to explain why the alpha and beta bands desynchronization is stronger in the right hemisphere of dysphagic patients compared to controls.

Another interesting result is that the desynchronization in control and healthy groups was not lateralized in the beta band; it was evenly distributed (LI close to zero) over the sensorimotor areas (C3, Cz, C4). The bilaterality of the healthy swallow has been previously described by the EEG study of Koganemaru et al. [19] and using other modalities like fMRI and NIRS [37,38], respectively. Furthermore, Hamdy et al. (1996) demonstrated that the swallowing muscles have cortical representations in both hemispheres and that each person may have a dominant hemisphere for swallowing (independent of manual laterality) [39]. In contrast, dysphagia patients exhibited a lateralized swallow control in alpha and beta bands. Similarly, the recent study by Ma et al. [40] compared the hemodynamic changes during volitional swallowing in unilateral post-stroke dysphagia patients, and a lateralized activation was also obtained. However, in the present work, although the LI confirms a general imbalance, there is no consistent bias towards either C3 or C4 within the dysphagic group, nor to the contralateral side of the stroke (results presented in Appendix A). Liu et al. presented similar results, with dysphagic patients not showing differences between left and right hemisphere activity [41]. Summing up, our findings support that bilateral control plays a crucial role in regulating swallowing, and that the dysphagic lateralization is not systematically directed towards one specific anatomical side, nor imposed by the laterality of the stroke.

This study is not without limitations that will be addressed in future research. Despite the study’s emphasis on comparing dysphagic patients and controls, which exhibited remarkably similar age distributions, the healthy subject group demonstrated significant differences in this regard. These variations could have potentially influenced the comparisons between stroke groups and healthy subjects. The results should be interpreted as exploratory, and further validation with larger samples and confirmatory analyses with multiple corrections would be necessary. The relationship between the observed neural patterns and clinical variables (such as stroke localization, dysphagia severity, etc.) was not explored. Further experimental studies—possibly involving complementary neuroimaging modalities or intervention-based paradigms—would be valuable to explore the underlying electrophysiological mechanisms in greater depth. Additionally, a larger sample size is required to enhance the generalizability of the findings and to facilitate a more robust analysis of the combined effects of stroke location, hemispheric lateralization, severity, and age on swallowing-related neural activity. It is worth noting, however, that the sample size in the present study exceeds that of most comparable studies in the field.

## 5. Conclusions

The use of ERSP analysis of EEG has enabled the characterization of dysphagic swallowing-related neural dynamics. This study identified two EEG-derived biomarkers with potential clinical value: (i) reduced alpha and beta desynchronization at C3 and Cz in dysphagic patients, which may reflect impaired somatosensory feedback integration and motor planning within the left sensorimotor cortex; and (ii) abnormal lateralization, indicating disrupted interhemispheric balance in swallowing control. These alterations point to cortical hypoactivation and asymmetric recruitment as physiological correlates of post-stroke dysphagia. Such biomarkers not only provide objective signatures that distinguish dysphagic from non-dysphagic patients but also hold promise for longitudinal monitoring of rehabilitation outcomes. Given its accessibility, EEG could thus complement gold-standard imaging tools and support more individualized and cost-effective management of dysphagia.

## Figures and Tables

**Figure 1 sensors-25-06767-f001:**
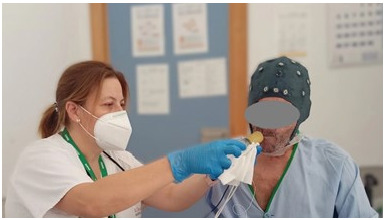
Photo of the recording setup and electrode placement.

**Figure 2 sensors-25-06767-f002:**
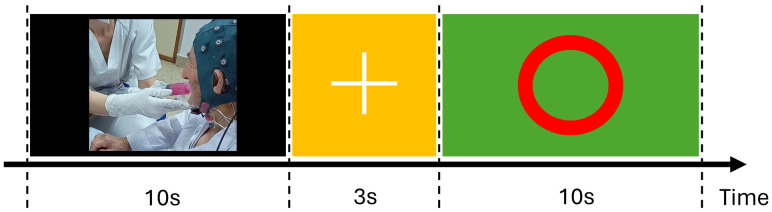
Swallowing protocol. Black screen: bolus given via syringe; yellow screen with cross: subject prepares; green screen with red circle: swallow bolus.

**Figure 3 sensors-25-06767-f003:**
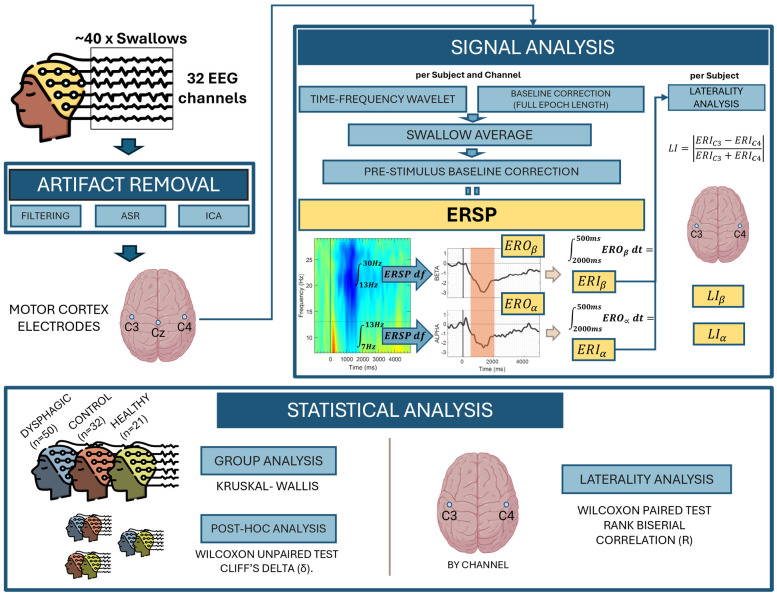
Schematic representation of the methodology pipeline, which includes preprocessing, analysis, characterization, and statistical analysis.

**Figure 4 sensors-25-06767-f004:**
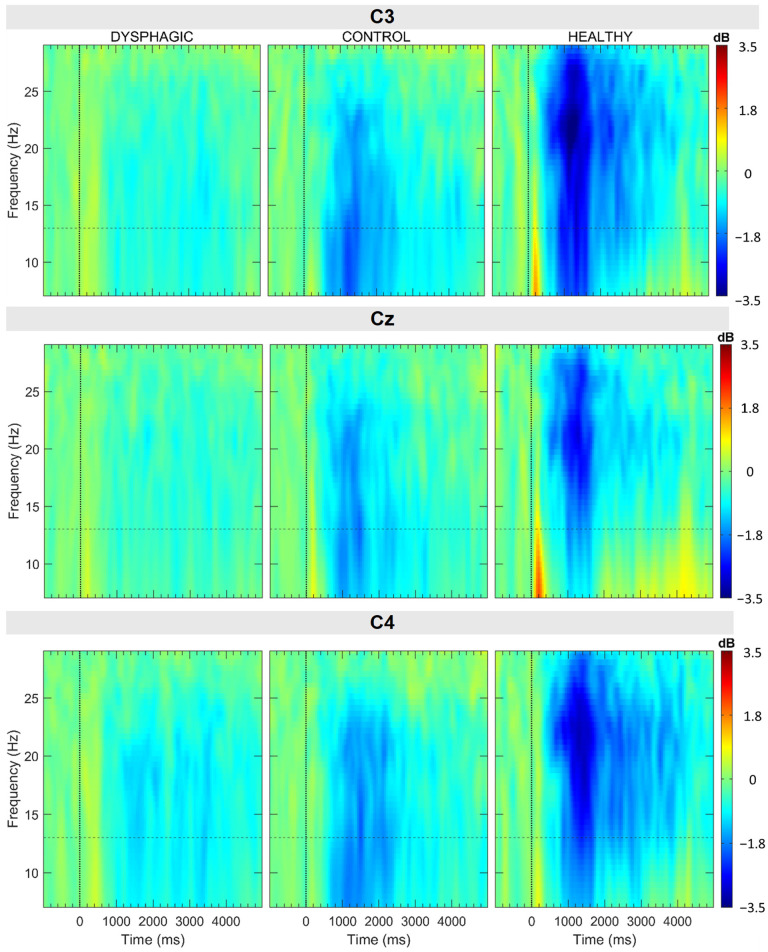
Grand average ERSP of the C3, C4, and Cz channels represented over the whole swallow (−1000 ms to 5000 ms) and the bandwidth (7 Hz to 30 Hz) within each group. A dashed horizontal line delimits alpha and beta bands.

**Figure 5 sensors-25-06767-f005:**
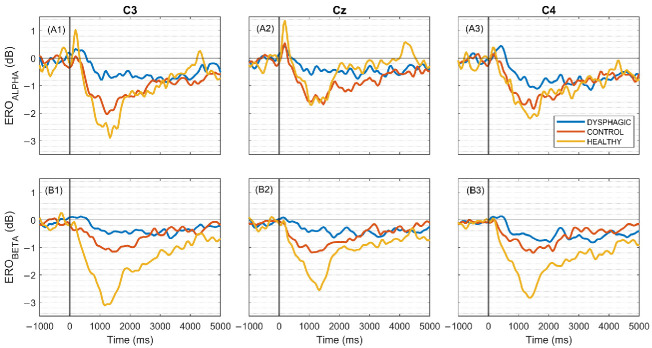
At the upper row, grand average ERO of each group for C3 (**A1**), Cz (**A2**), and C4 (**A3**) integrated across the alpha band (7–13 Hz). At the lower row, the same for beta band (13–30 Hz) for (**B1**) C3, (**B2**) Cz, and (**B3**) C4. These figures illustrate the time–frequency dynamics of event-related activity within each frequency range. The instruction of the swallow is denoted at 0 ms by a vertical line. Healthy:* n* = 21 subjects, mean 39.95 ± std 0.22 swallows/subject; Dysphagic: *n* = 47 subjects, 33.50 ± 8.65 swallows/subject; Control:* n* = 31 subjects, 35.41 ± 7.69 swallows/subject.

**Figure 6 sensors-25-06767-f006:**
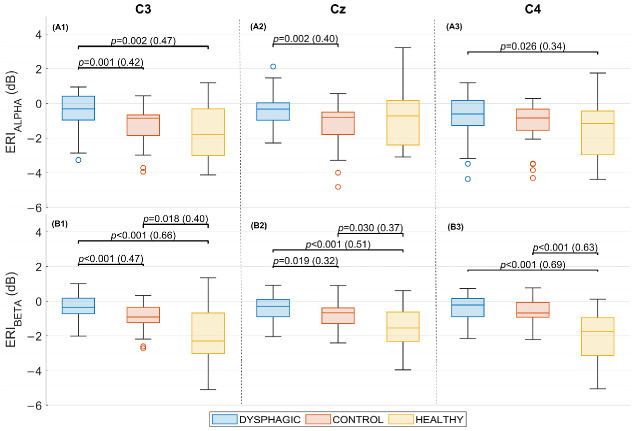
ERI in the alpha band (7–13 Hz), upper row, and beta band (13–30 Hz), lower row, for each electrode over the time window of maximum differences (500 ms to 2000 ms) for each group. Panels correspond to: (**A1**) C3 alpha band, (**A2**) Cz alpha band, (**A3**) C4 alpha band; and (**B1**) C3 beta band, (**B2**) Cz beta band, (**B3**) C4 beta band. Statistically significant differences, *p*-value, and size effect (δ) between paired groups are indicated.

**Figure 7 sensors-25-06767-f007:**
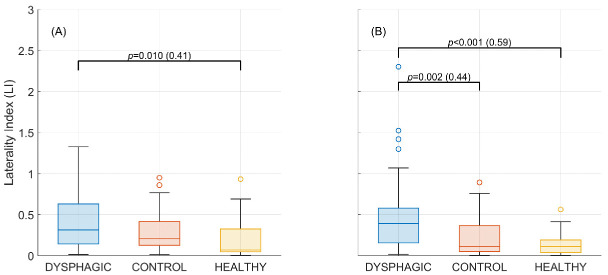
Lateralization index during the interval of maximum differences (500 ms to 2000 ms) per group over the (**A**) alpha band (7–13 Hz) and (**B**) beta band (13–30 Hz). Statistically significant differences, *p*-value, and size effect (δ) between paired groups are indicated.

**Table 1 sensors-25-06767-t001:** Relevant demographic variables (age, gender, and stroke laterality) from the different study groups (Healthy, Control, and Dysphagic).

Characteristic	Healthy	Control	Dysphagic	*p*-Value
Age [years] (mean ± std)	59.9 ± 6.3	67.1 ± 11.4	69.0 ± 12.9	*p* = 0.014 (ANOVA)
Gender [*n* (%)]:				*p* = 0.5532 (Chi^2^)
Masculine	11 (52.38%)	17 (53.1%)	33 (66.0%)	
Feminine	10 (47.62%)	15(46.9%)	17 (34.0%)	
Stroke Laterality [*n* (%)]:				*p* = 0.8392 (Chi^2^)
Right	-	18 (56.3%)	27 (54%)	
Left	-	13 (40.6%)	20 (40%)	
Bilateral	-	1 (3.1%)	3 (6%)	

**Table 2 sensors-25-06767-t002:** Statistical comparison: *p*-value (effect size (r)) for ERI paired C3 vs. C4 electrode comparisons within each group and frequency band. Blue cells indicate significance (*p* < 0.05).

ERI	C3 vs. C4
Dysphagic (alpha)	0.102 (r = 0.23)
Dysphagic (beta)	0.282 (r = 0.00)
Control (alpha)	0.295 (r = −0.19)
Control (beta)	0.003 (r = −0.65)
Healthy (alpha)	0.639 (r = −0.10)
Healthy (beta)	0.370 (r = 0.10)

## Data Availability

The datasets presented in this article are not readily available because the data are part of an ongoing study. However, data supporting the results presented in the manuscript are available upon reasonable request.

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
