# Peer review of "Electroencephalographic Characterization of Sensorimotor Neural Activation During Swallowing in Dysphagic Patients"

_sensors, 2025, doi:10.3390/s25216767_

Round 1

Reviewer 1 Report

Comments and Suggestions for Authors

This is an interesting preliminary study addressing an often overlooked medical condition. Overall I think the manuscript presents some interesting results and could certainly foster future research in this direction. The manuscript is well-written for the most part, with additional clarifications needed here and there. I have one major concern regarding potential confounding factors, which I believe could be addressed better on the current sample (see below). Nevertheless, limitation are acknowledged in the discussion.

My main issue is with how laterality of the stroke is handled in the analytical framework. Towards the end of the discussion, the Authors state "The risk of interpretational bias due to lesion lateralization is minimized" as the laterality (left/right or bilateral) subgroups are roughly even in the study groups (not confirmed via Chi-square test). I believe this could be much better addressed if instead of the scalp location C3-Cz-C4 framework the data is instead analyzed in an ipsilateral-central-contralateral manner, where C3/C4 electrodes are sorted into the appropriate group given the stroke laterality of the participant. No significant effect of stoke lateralization could confirm the claim that it has no effect on the neural signature.

Also, the characteristics of the different study groups (such as laterality of stroke) should be reported at Section 2.1, instead of at the end of the Discussion. It should also be confirmed that the different study groups did not differ in terms of age, distr. of sex, distr. of stoke lateralization, or any other relevant demographic variable that might potentially influence the results (e.g., handedness).

The introduction mentions relatively low detection rates for dysphagia (64-78%), "indicating a high likelihood of underdiagnosis.". How does this affect the certainty of group assignment in the present study sample e.g., is there a concern that dysphagia was underdiagnosed in the control group(s), or false positive diagnoses in the dysphagia group? Can/should this be addressed?

At line 161 describing data selection, I would highlight that t=0 reflects the red circle cue (and not the cross cue) to avoid any confusion.

Were assumptions of the utilized statistical tests verified? Were post hoc analyses adjusted for multiple comparisons? Why not utilize a two-way rm ANOVA model with group as between-subject effect and electrode site as within-subject effect? It would also be important to report effect sizes for the observed outcomes.

In the discussion: "The key contribution of this study lies in the identification of significant differences in Event-Related Spectral Perturbation (ERSP) patterns between individuals with dysphagia and healthy control subjects." I would disagree; it is more relevant to show differences between stroke patients with- and without dysphagia (comparing with healthy controls cannot attribute the difference to dysphagia instead of stroke itself).

line 346-347: I would remove the comment regarding the corpus callosum. EEG primarily measures the activity of layer V pyramidal cells and not white matter tracts (which comprise the corpus callosum). Interhemispheric communication could be better assessed via C3/C4 (or ipsi-/contralateral) connectivity analysis. 

Lines 351-352: this should be moved to the Methods, and should be confirmed that the proportions in the relevant study groups are comparable.

Author Response

We thank the reviewer for their valuable feedback and hope that the changes made have enhanced the overall quality of the revised manuscript. The specific modifications are detailed in the document attached.

Reviewer 2 Report

Comments and Suggestions for Authors

This paper has significant issues in both experimental analysis and writing quality.

First, regarding the experimental section: although the paper has collected and statistically analyzed data, it lacks in-depth exploration of the reasons behind the data results. Meanwhile, the paper fails to effectively compare its findings with those of previous relevant studies and also lacks the demonstration of quantitative numerical experiments. On the whole, the experiment reads more like a report on data collection—it lacks in-depth reflection on the data and shows insufficient innovation.

Second, the paper’s writing quality is poor. For instance, there are errors in formula numbering, inconsistencies in paragraph indentation and capitalization, and numerous unnecessary blank spaces in formatting.

Therefore, further revisions are required.

Author Response

We thank the reviewer for their feedback and hope that the changes made have enhanced the overall quality of the revised manuscript. The specific modifications are detailed in the point-to-point response document.

Reviewer 3 Report

Comments and Suggestions for Authors

This study investigated changes in alpha and beta event-related desynchronization (ERD) at C3, Cz, and C4, as well as hemispheric lateralization indices, in 50 post-stroke dysphagic patients, 30 post-stroke non-dysphagic controls, and 21 healthy adults during the performance of a swallowing task.

The manuscript was clearly written, and the methods were clearly explained. The results are relevant to the research field.

Minor comments:

  • In the introduction section, the authors claim: “This article presents the first ERSP characterization of EEG signals to identify biomarkers related to the strength and lateralization of activity in post-stroke dysphagia.” However, in the abstract section, the ERSP was not explicitly mentioned. Please clarify this point.
  • The relationship between ERSP and ERD should be more clearly explained in the introduction and methods sections.
  • Figure 3 legend should mention that the graphs represent a grand average.
  • Does Figure 4 also represent a grand average?
  • Figure 4 should include information about the number of samples employed per subject to obtain the averages and the total number of samples for all the subjects.

Author Response

We thank the reviewer for their valuable feedback and hope that the changes made have enhanced the overall quality of the revised manuscript. The specific modifications are detailed in the point-to-point response document.

Round 2

Reviewer 1 Report

Comments and Suggestions for Authors

I thank the Authors for addressing my comments. I believe that the manuscript now presents the analyses and their outcomes more comprehensively, also appropriately acknowledging potential study limitations.

Reviewer 2 Report

Comments and Suggestions for Authors

The author has addressed my concerns.